# Influence maximization in Boolean networks

Thomas Parmer [1], Luis M. Rocha [2,3] & Filippo Radicchi [1✉]

The optimization problem aiming at the identification of minimal sets of nodes able to drive the dynamics of Boolean networks toward desired long-term behaviors is central for some applications, as for example the detection of key therapeutic targets to control pathways in models of biological signaling and regulatory networks. Here, we develop a method to solve such an optimization problem taking inspiration from the well-studied problem of influence maximization for spreading processes in social networks. We validate the method on small gene regulatory networks whose dynamical landscapes are known by means of brute-force analysis. We then systematically study a large collection of gene regulatory networks. We find that for about 65% of the analyzed networks, the minimal driver sets contain less than 20% of their nodes.

[1] Center for Complex Networks and Systems Research, Luddy School of Informatics, Computing, and Engineering, Indiana University, Bloomington, IN 47408, USA. [2] Consortium for Social and Biomedical Complexity, Systems Science and Industrial Engineering Department, Thomas J. Watson College of Engineering and Applied Science, Binghamton University (State University of New York), Binghamton, NY 13902, USA. [3] Instituto Gulbenkian de Ciência, Oeiras 2780-156, Portugal. ✉email: filiradi@indiana.edu

Determining the influence of nodes in networks is critical for understanding and controlling real-world systems[1]. Applications include identifying super spreaders in marketing and political campaigns[2], immunization targets for disease containment[3], vulnerable nodes in financial networks[4], and key therapeutic targets in biological signaling and regulatory networks[5–7].

A large body of research within the network science literature focuses on the problem of influence maximization, i.e., identifying the nodes that maximize a spreading process on a network[8]. In the standard problem setting, all nodes in the network are initially set to an inactive state. A fixed number of seed nodes are activated to initiate a spreading process in the network. The influence of the seed set is measured in terms of the size of the outbreak, i.e., the total number of nodes activated during the spreading process. The size of the outbreak depends on the rules of the spreading process, the structure of the underlying network, and the nodes in the seed set. The problem of influence maximization is thus the identification of the seed set, among all possible with prescribed size, that generates the largest average outbreak in the network. This problem is known to be NP-hard for both simple and complex contagion processes, and thus exactly solvable only in extremely small networks[9]. Approximate but effective strategies to solve the influence maximization problem have computational complexity that ranges from cubic to linear[10].

Here, we consider the generalization of the influence maximization problem to Boolean networks, a type of discrete dynamical systems[11]. In a Boolean network, we are not necessarily interested in maximizing the number of nodes that are activated during the dynamics; rather, our goal is identifying sets of driver nodes that are sufficient to control the dynamical system toward some desired final configuration[12–14]. Note that some of the spreading models considered in the standard formulation of the influence maximization problem, e.g., the linear threshold model[15], can be seen as a special formulation of the more general family of problems that we consider here. The problem of identification of a driver set in Boolean networks is known to be polynomial for tree-like structures, but NP-hard for general networks[14]. One way in which we simplify the problem is by focusing on identifying the smallest driver set for a specific attractor[16,17]. Our minimization problem is analogous to the one generally considered in control theory for networks[18], but with the non-trivial goal of accounting for the nonlinearity characterizing Boolean dynamics.

We remark that Boolean networks can be mapped to discrete-time linear dynamical systems[19]. However, the exact optimization problem is computationally intractable due to the fact that the size of the resulting system is $2^N$, with $N$ equal to the number of nodes in the original Boolean network. One way to obtain approximate solutions in reasonable time relies on utilizing only the structure of the underlying Boolean network, by linearizing the nonlinear dynamical rules that regulate the evolution of the system[20]. Whereas linear structure-based control methods may be effective in some types of Boolean dynamics, they do not provide any guarantee of finding the best set of nodes to control a network[17].

Another way to approximate sufficient driver node sets is by identifying the nodes that break feedback vertex sets[21,22]. This method does not linearize dynamics and finds a driver set that can control the network to any attractor; however, it can only provide the driver set that controls the ensemble of all dynamical systems that fit the same network structure and has non-polynomial complexity[21,23] which hinders applicability to large networks. Other methods exist to find optimal driver sets toward specific attractors for a given Boolean network. For example,

Zañudo and Albert successively find partial fixed points of the network dynamics (called stable motifs) that guide the dynamics toward an attractor of interest[24]; Kim et al. use genetic algorithms and a network's attractor landscape to find the minimal driver set (called the control kernel) to guide the dynamics to a specified attractor[25]; Borriello and Daniels similarly find control kernels by pinning nodes to their specified state in the desired attractor that distinguish that particular attractor from other attractors[26]. Unfortunately, all of these methods have exponential complexity, in the general case.

Toward a feasible approximate method for identifying the nodes that control dynamics to a desired attractor, we first deploy an individual-based mean-field approximation (IBMFA) for Boolean network dynamics. As in the IBMFA used in the study of spreading processes on networks[27], our IBMFA also consists in neglecting dynamical correlations among variables so that every node integrates the average, over an infinite number of independent realizations of the dynamical process, behavior of its neighbors. Mean-field approaches to Boolean network dynamics exist, including the classic approach by Derrida and Pomeau[28] and other recent attempts, e.g., refs. [29–32]. Those attempts are developed for the so-called annealed networks, thus they are devised to deal with ensembles of networks, where the network structure and/or the rules of the Boolean dynamics are stochastic. The resulting approximations allow to describe the behavior of the dynamical system averaged over the given ensemble of networks. We differentiate from these previous attempts by developing an approach that is valid for quenched networks. Our approach takes as input a given network structure and prescribed Boolean rules, and generates as output the average trajectory of the dynamical system started from stochastic initial configurations. We show that our IBMFA accurately reproduces the average dynamical behavior estimated from numerical simulations of individual node states in both random Boolean networks (RBNs) and gene regulatory networks (GRNs).

Second, we introduce a statistical notion of control or influence for Boolean dynamics. Accordingly, the influence of a set of nodes is quantified in terms of the entropy associated to long-term configurations reachable by the system when controlled by that specific set of nodes but otherwise started from a maximally uncertain initial condition. We construct optimal sets of influential nodes by means of greedy optimization[33]. The algorithm scales cubically with the network size, thus allowing the analysis of systems that cannot be studied with brute-force approaches. The algorithm is used to approximate the minimum-size driver set required to reach a known attractor by simply constraining the search over configurations compatible with the attractor. Also, the algorithm is used in unconstrained searches for optimal sets of nodes able to drive the system toward an attractor with a large basin. We validate our method on known attractors of the *Drosophila melanogaster* segment polarity single-cell and para-segment networks[34]. Also, we recover known effects of anti-cancer drugs on the estrogen receptor breast cancer network [35], and find minimal driver sets in the networks representing the yeast *Saccharomyces cerevisiae* cell-cycle[36] and the T cell large granular lymphocyte leukemia[37]. We then systematically apply our method to large collections of synthetic and real networks. We find that the relative size of the optimal driver set in RBNs toward unspecified attractors increases as the degree of the network increases, but is invariant with the system size. GRNs within the Cell Collective repository[38] are also characterized by optimal driver sets whose relative size is independent of the system size. Our predictions for networks within the Cell Collective repository[38] are in very good agreement with those obtained by the method of Borriello and Daniels[26].

## Results

**Accuracy of the individual-based mean-field approximation.**
We consider arbitrary Boolean networks[39], see the Methods Section "Boolean networks" for details. The state of node $i$ at time $t$ is denoted by the binary-valued variable $\sigma_i(t) = 0, 1$, while a generic configuration of the system is denoted by $\vec{\sigma}(t) = [\sigma_1(t), \dots, \sigma_N(t)]$, with $N$ standing for the network size. The dynamics of node $i$ is specified by the lookup table $F_i$, whose inputs are the state of all its $k_i$ neighbors, see Eq. (1). Under synchronous updating, given the configuration $\vec{\sigma}(t)$, the system evolves in a deterministic fashion to another configuration $\vec{\sigma}(t+1)$. A full description of the system's dynamics can be given in terms of its associated state-transition graph (STG), where each node corresponds to one of the $2^N$ possible configurations and a single directed edge indicates the transition from one configuration to another. Understanding the dynamical properties of a Boolean network from its STG is straightforward. However, the very fact that the size of the STG grows exponentially with the system size limits its practical relevance to very small systems only.

To overcome this limitation, we deploy an individual-based mean-field approximation (IBMFA) aimed at describing the average behavior of nodes in the Boolean network, see Section "Individual-based mean-field approximation" for details. In the IBMFA, the dynamical state of node $i$ is represented by the real-valued variable $s_i(t)$, standing for the probability of finding the node active at stage $t$ of the dynamics, i.e., $s_i(t) = P(\sigma_i(t) = 1)$. In this model of the dynamics, each node is influenced only by the average behavior of its neighbors, i.e., dynamical correlations among variables are neglected, see Eq. (4). IBMFA requires to sum over all the entries of the lookup tables. Thus, unlike STG-based methods but similarly to causal graph methods[16,40], the approximation grows linearly with $N$ and exponentially with the degree of the nodes. As a result, IBMFA is feasible to compute in large networks as long as the degree of the nodes is not too large, making it applicable to many real-world sparse networks.

The average trajectory quantified by IBMFA is taken over realizations of the dynamical system and is conditioned by the distribution of the initial configuration $P(\vec{\sigma}(t=0) | \vec{s}(t=0))$, see Eq. (3). As a result, if the initial condition is certain, i.e., $s_i(0) = 0, 1$ for all $i$, then IBMFA reproduces the ground-truth trajectory on the STG started from the given initial configuration. Also, no average is taken on either the network structure or the Boolean lookup tables. This is the main point of differentiation between our IBMFA and existing mean-field approaches for Boolean networks[28–32].

In order to test the accuracy of our approximation, we compare IBMFA predictions with results from numerical simulations. Some results of our tests are shown in Fig. 1.

First, we test the accuracy of the IBMFA in random Boolean networks (RBNs), see Fig. 1a. We generate RBNs with $N = 100$. Each node has degree equal to $k$. The output of each of the $2^k$ rows of the lookup table $F_i$ of node $i$ is set equal to either 0 or 1 with identical probability. The results of Fig. 1a correspond to averages taken over 100 independent RBNs for each $k$ value. Initial configurations of the dynamics are randomly generated according to the probability of Eq. (3), where we set $s_i(t=0) = 1/2$ for all nodes $i$. We sample $R = 100$ random initial configurations per network instance. We measure the mean squared error of the IBMFA prediction with respect to the ground-truth average trajectory estimated from numerical simulations, see Eq. (5). For $k > 1$, the error begins to plateau to a non-null value at $t \simeq 10$; furthermore, the error decreases by increasing the degree from $k = 2$ to $k = 3$. For $k = 1$, the error quickly goes to zero. This is due to the peculiar ring structure of the graph. Because our goal is

to determine influence to long-term configurations independent of the starting conditions, it is important that the IBMFA accurately reproduces the behavior of the system averaged over all possible configurations. Empirically, an observer may only measure one instance of a network's dynamics which may not be very indicative of the average dynamical behavior. The comparison with the variance of the sample of configurations used to estimate the ground-truth average trajectory, see Eq. (6), tells us that IBMFA is more informative than observing the outcome of a single instance of the network dynamics if the goal is to predict the average behavior of the system.

Also, we test IBMFA on gene regulatory networks (GRNs) and other biological signaling networks including the single-cell *Drosophila melanogaster* segment polarity network (SPN), see Fig. 1b. The network is composed of $N = 17$ nodes only, thus still approachable by means of the brute-force STG analysis. Initial configurations of the dynamics are randomly generated according to the probability of Eq. (3), where we set $s_i(t=0) = 1/2$ for all nodes $i$. Results are obtained over $R = 100$ independent initial configurations of the dynamics. Also here, we see that the error reaches a non-zero plateau value for $t \simeq 10$. The plateau value of the baseline error is much larger than the one observed for IBMFA. Similar findings are obtained also for the networks representing the T cell large granular lymphocyte (T-LGL) leukemia ($N = 60$) and the estrogen receptor (ER+) breast cancer ($N = 80$).

In the Supplementary Information (SI), we repeat the analysis for different updating schemes, including deterministic asynchronous, stochastic asynchronous, and block deterministic updating schemes[41–43]. We find that the evolution of the mean squared error associated with the IBMFA is influenced by the specific updating rule considered, see Fig. S1. For the *Drosophila melanogaster* SPN we find that the long-term value of the IBMFA error is almost insensitive to the updating scheme used. Instead, for the yeast cell-cycle network, we observe more apparent differences between IBMFA errors depending on the updating scheme at hand.

**Dynamical influence of nodes.** We study how some externally controlled nodes affect the dynamical behavior of a Boolean network. Based on the analogy with problems considered in the context of spreading processes on social networks[8,9], we use the term seed for a node that is externally controlled, and the term influence to indicate the effect of one or more seeds on the dynamics of the network. When the former have maximum influence, they are also known as driver nodes in the context of controllability.

We denote a generic set of seed nodes as $\mathcal{X}$, see Section "Definition of seed set". Strictly speaking, $\mathcal{X}$ is a set of tuples of the type $(i, \hat{\sigma}_i)$, each specifying the label of the nodes belonging to the set as well as their imposed state value. We say that node $i$ belongs to $\mathcal{X}$, i.e., $i \in \mathcal{X}$, if the label of node $i$ appears in one of the tuples of the set. Please note that node $i$ may appear at maximum in one tuple of $\mathcal{X}$, as either $(i, \hat{\sigma}_i = 0)$ or $(i, \hat{\sigma}_i = 1)$. Nodes in the set $\mathcal{X}$ can influence the dynamics of other nodes that do not belong to $\mathcal{X}$; we assume, however, that seeds do not change their state during the dynamics of the system, i.e., $\sigma_i(t) = \hat{\sigma}_i$ for all $i \in \mathcal{X}$ and for all $t \geq 0$. This is known in the literature as pinning control[17,21,23]. The assumption is identical to the one underlying the study of influence in irreversible spreading processes on social networks[9]. Here, the invariance of the dynamical state of the nodes in $\mathcal{X}$ serves to model systems whose typical time scale is much shorter than the one used to perturb the state of the seed nodes. This is a good assumption for some applications, as for example the study of the effect of drugs in GRN dynamics.

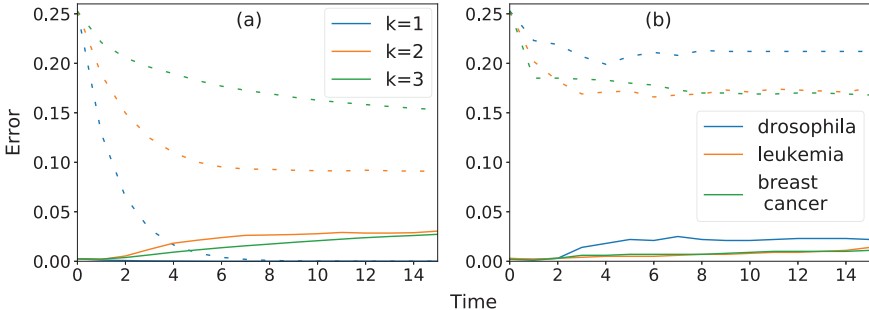

**Fig. 1 Accuracy of the individual-based mean-field approximation. a** We evaluate the mean squared error $e(t)$, as defined in Eq. (5), of the prediction made under the individual-based mean-field approximation (IBMFA) with respect to the ground-truth configuration of the system at stage $t$ of the dynamics. The systems under observation are random Boolean networks (RBNs) with $N = 100$ nodes and fixed degree $k$. Initial configurations are sampled from Eq. (3), where the probability of node $i$ to be active equals $s_i(t=0) = 1/2$ for all $i = 1, ..., N$. We consider 100 RBNs for each value of $k$. For each of them, ground-truth trajectories used in the computation of the IBMFA error are estimated by relying on $R = 100$ independent simulations. We then display the average error over the 100 RBNs using full curves of different colors depending on the degree $k$ of the underlying RBN. IBMFA errors are compared to their corresponding baseline values $b(t)$, i.e., the variance of the sampled trajectories used to estimate the average trajectory of the network [see Eq. (6)], which are shown as dashed curves in the plot. **b** Same as in panel a, but for selected biological signaling networks: the *Drosophila melanogaster* segment polarity network (drosophila), the T cell large granular lymphocyte leukemia network (leukemia), and the estrogen receptor breast cancer network (breast cancer). Initial configurations are sampled from Eq. (1), where the probability of node $i$ to be active equals to $s_i(t=0) = 1/2$ for all $i = 1, ..., N$.

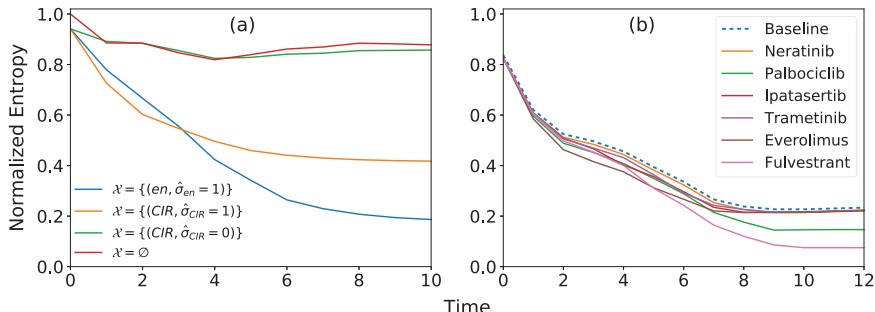

**Fig. 2 Dynamical influence in Boolean networks. a** We monitor the residual entropy of the *Drosophila melanogaster* segment polarity network as a function of time. Entropy is measured in bits and is normalized by the size of the network. Different curves correspond to different choices for the set of seed nodes $\mathcal{X}$. Specifically, we report results for $\mathcal{X} = \emptyset$, $\mathcal{X} = \{(en, \hat{\sigma}_{en} = 1)\}$, $\mathcal{X} = \{(CIR, \hat{\sigma}_{CIR} = 1)\}$, and $\mathcal{X} = \{(CIR, \hat{\sigma}_{CIR} = 0)\}$. **b** Same as in panel a, but for the estrogen receptor breast cancer network. The baseline is given by a seed set where 12 nodes of the network are set to biologically relevant states and the drug *Alpelisib* is activated, i.e., $\mathcal{X} = \{(SGK1\_T, \hat{\sigma}_{SGK1\_T} = 0), (IGF1R\_T, \hat{\sigma}_{IGF1R\_T} = 1), (PIM, \hat{\sigma}_{PIM} = 0), (ER, \hat{\sigma}_{ER} = 1), (PBX1, \hat{\sigma}_{PBX1} = 1), (HER2, \hat{\sigma}_{HER2} = 0), (HER3\_T, \hat{\sigma}_{HER3\_T} = 0), (PTEN, \hat{\sigma}_{PTEN} = 0), (BIM\_T, \hat{\sigma}_{BIM\_T} = 0), (BCL2\_T, \hat{\sigma}_{BCL2\_T} = 0), (PDK1, \hat{\sigma}_{PDK1} = 0), (mTORC2, \hat{\sigma}_{mTORC2} = 0), (Alpelisib, \hat{\sigma}_{Alpelisib} = 1)\}$[35]. The various curves are obtained by activating a specific anti-cancer drug (other than *Alpelisib*) and adding it to the baseline.

We assess the influence of the set of seeds $\mathcal{X}$ in a Boolean network in terms of the residual uncertainty about the states of other nodes that do not belong to $\mathcal{X}$. It is measured assuming that the state of the nodes in $\mathcal{X}$ is known and does not change during the dynamics. The notion is similar to the one used in ref. [44]. Specifically, we assume that the initial configuration is randomly sampled from the distribution of Eq. (1). In doing so, the state of the nodes in $\mathcal{X}$ is set deterministically, i.e., $s_i(t=0) = \hat{\sigma}_i$ for all $i \in \mathcal{X}$. By contrast, we have maximal uncertainty for all other nodes, i.e., $s_i(t=0) = 1/2$ for $i \notin \mathcal{X}$. We measure the residual uncertainty of the system at time $t$ as the entropy of the probability distribution of the configurations reachable by the system at time $t$, see Eq. (7), conditioned to the known state of the nodes in $\mathcal{X}$. The dynamical influence of the set $\mathcal{X}$ is inversely proportional to the long-term residual uncertainty of the system.

To speed up the computation of the entropy, we rely on the IBMFA. The approximation provides us with a state probability value $s_i(t)$ for each node $i$ in the network, and such a value can be readily plugged into Eq. (7) for the computation of the entropy. The use of IBMFA is justified by a good level of agreement, both at the level of individual nodes and configurations, with the

ground-truth entropy estimates from numerical simulations (see Figs. S2 and S3 for details).

We monitor the entropy of various Boolean networks conditioned by different seed sets $\mathcal{X}$. We find that different sets of nodes can have very different dynamical influence on the network. The effect strongly depends not just on what nodes are in the set but also on the state imposed on these nodes.

In Fig. 2a for example, we show how the entropy of the *Drosophila melanogaster* SPN evolves in time. We consider different choices for the set $\mathcal{X}$. Clearly, maximal initial uncertainty is present for $\mathcal{X} = \emptyset$. Such an uncertainty typically decreases as the system dynamics evolves. We note that some nodes have more influence than others in reducing the residual entropy of the system. For example, the set $\mathcal{X} = \{(en, \hat{\sigma}_{en} = 1)\}$ has more dynamical influence on the network than the set $\mathcal{X} = \{(CIR, \hat{\sigma}_{CIR} = 1)\}$. Also, it is important to stress that dynamical influence is due not just to the identity of the nodes, but also to their imposed state. For example, $\mathcal{X} = \{(CIR, \hat{\sigma}_{CIR} = 1)\}$ has dynamical influence larger than $\mathcal{X} = \{(CIR, \hat{\sigma}_{CIR} = 0)\}$. We note that in all cases, entropy values plateau after $t \simeq 10$ stages of the dynamics. We treat this value as representative for the long-term behavior of the system, and use it in our operative definition of long-term dynamical influence of a

seed set. We find that entropy values similarly plateau after $t \simeq 10$ in other biological networks and, importantly, that there is little or no change in rank order of seed sets after this value. We, therefore, use this value in our analysis of all biological networks unless stated otherwise.

The above considerations are confirmed if network dynamics evolve according to different updating schemes, see Figs. S4–S5. System entropy follows a trajectory that depends on the specific updating scheme considered, however, its long-term value is almost the same irrespective of the updating scheme at hand for the *Drosophila melanogaster* SPN. Some differences appear for the yeast cell-cycle network.

We further study systematically the influence of all possible $2^{|\mathcal{X}|}$ ($0 exN |\mathcal{X}|$) seed sets of size $|\mathcal{X}| \leq 3$ for the *Drosophila melanogaster* SPN and the T-LGL leukemia network (see Figs. S6 and S7). For the *Drosophila melanogaster* SPN, we find that three sets of size $|\mathcal{X}| = 3$ are able to reduce the long-term residual entropy to zero; none of the sets of size $|\mathcal{X}| < 3$ results in a null residual entropy. These results are consistent with the known control portrait of this network[17]. As a comparison, for the larger T-LGL leukemia network, no single seed set of size $|\mathcal{X}| \leq 3$ leads to residual entropy equal to zero.

For the ER+ breast cancer network, consistent with the original goals of the model[35], we focus our analysis on the dynamical influence of the input nodes representing anti-cancer drugs (Fig. 2b). Previous literature has shown that some drugs affect cell apoptosis or proliferation more than others[35,40]. With our approach, we reproduce those results. Specifically, we consider only biologically relevant configurations of the ER+ breast cancer cell baseline where 12 nodes representing biochemical components involved in signaling pathways are set to a particular state and the drug *Alpelisib*, i.e., an PI3K inhibitor, is active[35]. We then study the effect that the activation of an additional drug, as a model of multi-drug therapy on PIK3CA-mutant breast cancer cells, has on the long-term behavior of the dynamical system in order to determine which drugs best synergize with *Alpelisib*. As already shown by ref. [40], we find that only two drugs have significant impact on system dynamics in this context: *Palbociclib* and *Fulvestrant*. With our method, the impact of the two drugs appears as a reduction in the value of the long-term entropy of the network; all other drugs do not reduce entropy beyond the baseline, as shown in Fig 2b. This is consistent with prior causal analysis of this network, which has shown that only these two drugs affect additional signaling pathways not controlled by the baseline cell configuration, thereby having a synergetic effect with *Alpelisib* on apoptosis and proliferation of cancer cells in this model[40]. Thus our new approximate model reproduces the causal behavior of the network.

**Influence maximization**. The results of the above sections demonstrate that the IBMFA is effective in approximating the entropy of Boolean networks. Also, the entropy of the system conditioned by a seed set $\mathcal{X}$ is a meaningful quantity to assess the influence of the set $\mathcal{X}$ on the long-term dynamical behavior of the network. We leverage these results to develop an efficient algorithm (see Section "Influence maximization") for the identification of optimal sets of influential nodes in Boolean networks.

The algorithm constructs quasi-optimal sets of seeds with a greedy strategy, see Eq. (8). At each stage of the algorithm, the node whose control leads to the largest drop in the entropy function is added to the set. The algorithm has a known performance bound[33], and typically provides the best solution to several discrete optimization problems, as for example influence maximization in social networks[9,10].

A driver set is identified when the entropy reaches a null value, reflecting the fact that the long-term configuration of the system is fully determined by imposing the specified state of the nodes in the driver set. We stress that the set obtained at the end of the algorithm is not necessarily the optimal one. We, therefore, refine the identified driver set with a post-processing technique consisting in removing from the set all nodes that do not lead to an increase of the entropy of the system. Similar post-processing techniques are used to refine solutions to other discrete optimization problems[45,46].

Greedy selection can be used to find the minimal driver set to reach a given attractor or, if no constraints are specified, the minimal set to reach an attractor in the network, see Section "Influence maximization". This attractor is in a sense the easiest one to find via greedy selection and likely corresponds to the attractor with the largest basin.

We apply the algorithm to the *Drosophila melanogaster* SPN, see Fig. 3. The network is known to have 10 different attractors. By constraining greedy selection to target each of these attractors, we are able to find their corresponding optimal driver sets (Fig. 3b). Please note that post-processing the driver sets is required to reduce their size significantly without compromising the quality of the solution obtained (Fig. 3c). The minimal driver sets identified by our algorithm well approximate the ground-truth driver sets of the network obtainable by a brute-force exploration of the STG. Our predictions recover exactly the minimal driver sets for 7 of the 10 fixed points. We overestimate the size of the driver sets required to reach attractors 5, 9, and 10 by one node only (the PTC node). We note that overestimating the size of the ground-truth driver set by a small margin is reasonable given the level of approximation used by our approach, i.e., the IBMFA neglects dynamical correlations and the greedy optimization strategy is sub-optimal. We note that by definition the constrained versions of the algorithm display higher levels of uncertainty compared to the unconstrained version at time $t = 1$; however, due to the sub-optimality of our greedy algorithm, a smaller constrained driver set is found (attractor 1) than the one chosen by our unconstrained algorithm (attractor 4) prior to post-processing. Appropriately selecting seeds dramatically reduces the uncertainty of the system. The control by the best seed leads to a 60% reduction of the system entropy. By contrast, selecting random seeds has a mild effect on the long-term behavior of the dynamics, and entropy does not vanish even if 10 nodes are controlled.

Similar findings are valid for other Boolean network models of biochemical regulation and signaling, see Fig. S8. For the yeast cell-cycle network, we identify all optimal sets of seeds that control the system toward its 11 attractors[36]. We find that the size of the optimal seed sets is at least 4, consistent with what is known about the system[17]. When compared with the ground truth (see Fig. S9 and Table S1), we see that our method correctly retrieves the minimal driver sets for 4 out the 11 attractors of the yeast cell-cycle network, overestimates the driver sets by one or two nodes for 5 fixed points, and it underestimates the driver set by one node for 2 attractors. We recognize that underestimating the size of the driver sets is not a desirable outcome. On the other hand, we note that our underestimations are still good approximations of the ground truth, in the sense that the solutions found by our approach lead to the right attractors in 93% and 97% of the initial configurations, respectively. In other words, the additional node that is indeed required to always reach the desired fixed points is used only in respectively 7% and 3% of the total possible initial configurations. Also, we find that 6 biologically meaningful attractors of the *Drosophila melanogaster* parasegment network [34] can be reached by controlling no more than 11 nodes of the network, corresponding to 18% of the nodes

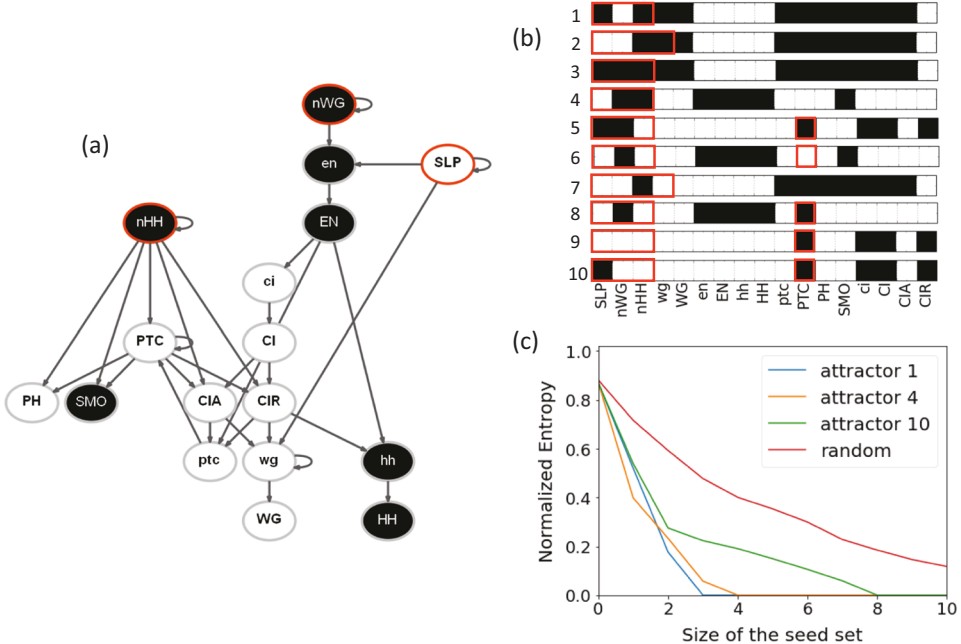

**Fig. 3 Driving Boolean networks to the desired attractor. a** We consider the *Drosophila melanogaster* segment polarity network. In the visualization, a directed connection from node $i$ to node $j$ indicates that the state of node $i$ is one argument of the Boolean function $F_j$ that regulates the dynamical evolution of node $j$. In the visualization, we represent one of the attractors of the network. Active nodes are represented in black; inactive nodes are depicted in white. The attractor may be reached by controlling the state of the three nodes highlighted in red. We identified the seed set using our algorithm; the seed set is $\mathcal{X} = \{(\text{SLP}, \hat{\sigma}_{\text{SLP}} = 0), (\text{nWG}, \hat{\sigma}_{\text{nWG}} = 1), (\text{nHH}, \hat{\sigma}_{\text{nHH}} = 1)\}$. **b** The 10 attractors of the network, and their corresponding minimal driving sets as identified by our algorithm. Attractor 4 is the same as in panel a. Our predictions recover exactly the minimal driver sets for 7 of the 10 fixed points. We overstimate the size of the driver sets required to reach attractors 5, 9, and 10 by one node only (the PTC node). **c** Residual entropy as a function of size of the seed set for three selected seed sets leading to specific attractors. The labels of the attractors are the same as in panel **b**. The unconstrained greedy selection process finds attractor 4. As a term of comparison, we display also the curve corresponding to seed sets composed of randomly selected nodes' indices/states. The curve displays the value of the entropy averaged over 100 random seed sets. All entropy values are estimated over $R = 100$ simulations of the dynamical process.

in the network. For example, the wildtype attractor can be reached by pinning only 10 nodes, which is a lower estimate than found by previous methods[16,23]. In particular, the entropy of the system displays a eight-fold reduction after the top 4 seeds are selected by the greedy algorithm for each of the 6 attractors. Additionally, we find that the T-LGL leukemia network[37] quickly reduces in entropy after the selection of the top 3 nodes via the greedy algorithm, and the network can be controlled with only 9 nodes (15% of the network size).

We apply systematically the greedy optimization algorithm to RBNs. Results are shown in Fig. 4a. No constraints are imposed in the search for the top influential nodes. We find that the size of the optimal driver set relative to the system size is a constant that depends on the degree of the graph only. We note that unbiased RBNs with homogenous degree $k \geq 2$ are in the chaotic regime and thus among the most difficult types of Boolean networks to control. In this respect, the results of Fig. 4a provide us with an upper bound of the size of the minimal driver set required to control homogeneous networks with given size and average degree. Introducing a bias in the output of the Boolean functions makes the network more controllable than an unbiased RBN, thus leading to a reduction in the size of the minimal driver set (see Fig. S10). This is due to the fact that biased networks are easy to control toward their biased attractors. However, controlling them towards other attractors may be more difficult.

Also, we systematically apply our unconstrained optimization algorithm to the 74 networks of the Cell Collective repository[38]. We measure node influence after $t = 10$ iterations of the IBMFA on some networks and verified that the value well represents the entropy values of the long-term dynamics of the system

(see Fig. S11). The results of Fig. 4b indicate that there is no apparent correlation (Pearson's $R = -0.22$, $p = 0.06$) between the relative size of the optimal driver set and the size of the network. We verified also that the relative size of the driver set does not correlate well with other topological and dynamical features of the networks, such as average degree, average bias, and mean effective connectivity[16] (see Figs. S12 and S13). Based on the analysis of the entire Cell Collective corpus, we compute the probability of a node to belong to the minimal driver set conditioned on its in-/out-degree, see Fig. S14. As expected, we find that nodes with no inputs, i.e., in-degree equal to zero, are always part of the set of drivers as these nodes cannot be controlled if not via external input. For non-null in-degree values, we find that nodes with sufficiently large in-/out-degree are significantly more likely to be in the minimal driver set than nodes with small in-/ out-degree centrality. This fact indicates that topologically central nodes are likely to be part of the minimal driver set.

Overall, we find that optimal sets of drivers contain less than 30% of the nodes for more than 80% of the networks in the repository (Fig. 4c). We compare our predictions with those by Borriello and Daniels of the average size of the minimal driver sets to fixed point attractors of a network[26]. Due to the high computational complexity of having prior knowledge of the fixed points of the network, comparisons are possible only on networks of relatively small size. As the results of Fig. 4d show, we find excellent agreement between the average size of driver sets predicted by our method and the predictions by Borriello and Daniels. As additional validation, we verify that the minimum-size driver set obtained with our method via unconstrained

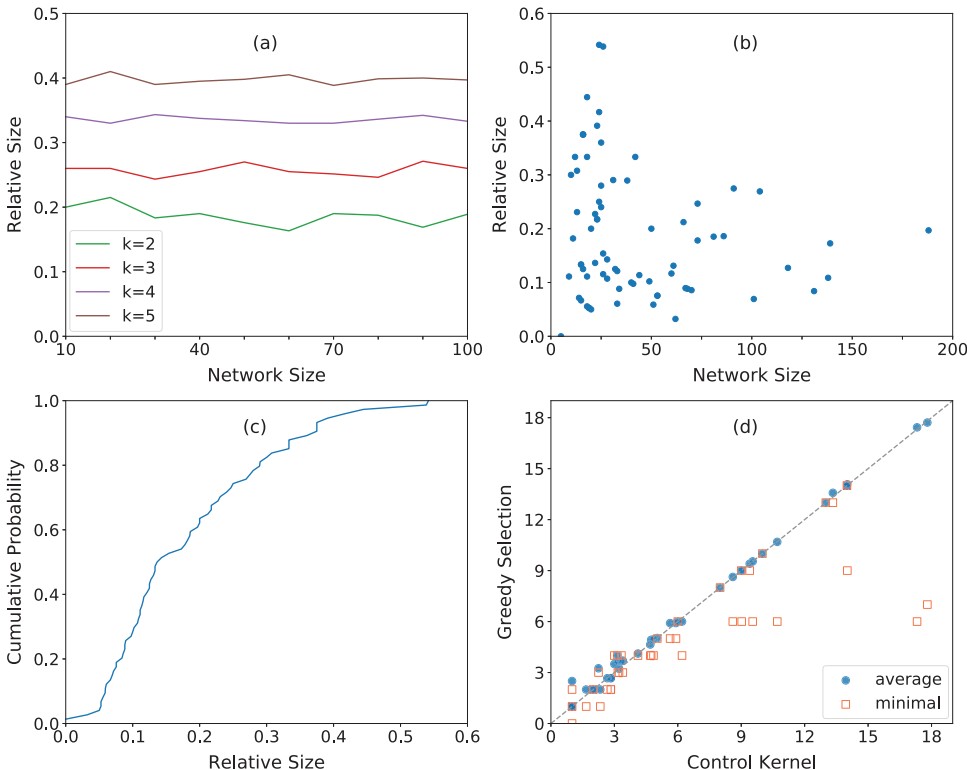

**Fig. 4 Influence maximization in synthetic and real Boolean networks. a** We identify the optimal sets of drivers by implementing our greedy strategy on random Boolean networks (RBNs) with variable network size $N$ and fixed degree $k$. We plot the relative size of the optimal driver set as a function of the network size. Different curves correspond to different $k$ values. Results refer to average values over 10 RBNs. **b** Relative size of the optimal driver sets identified by our algorithm in the 74 networks that are part of the Cell Collective repository. The relative size of the identified driver set is plotted against the network size of the corresponding network. **c** Cumulative distribution function of the relative size of the optimal driver sets of the Cell Collective networks shown in panel (**b**). **d** We compare our predictions with those obtained by ref. [26]. We consider networks from the Cell Collective repository that were analyzed in ref. [26] and have fixed point attractors, and plot the average size of the optimal driver sets predicted by our method vs. the average size of the control kernels (blue circles). A control kernel is an approximation of the minimal driver set towards a desired attractor[26]; optimal driver sets are instead estimated using our constrained optimization algorithm. Each point in the plot is a network. Average values are computed over all the possible fixed points of a network. The dashed gray line indicates an exact match between the two predictions. For the same set of networks, we visualize also the size of the minimal optimal driver set (computed using our unconstrained optimization algorithm) vs. the average size of the control kernels (orange squares).

optimization is consistently smaller than the average value of the minimal driver sets.

In the SI, we apply our unconstrained optimization algorithm in the search of the minimal driver set under the assumption that the system's dynamics is regulated by updating rules other than synchronous deterministic updates, see Fig. S15. For the *Drosophila melanogaster* SPN, outcomes of the analysis are almost identical to those valid for synchronous updating. However, results obtained for the yeast *Saccharomyces cerevisiae* cell-cycle network indicate that the long-term behavior of the network is quite sensitive to the specific updating scheme at hand. This latter observation is in line with the findings of ref. [41] regarding the change in the size of the basin of attraction of the fixed points depending on the updating scheme. We also apply constrained optimization toward given fixed points under the various updating schemes, see Figs. S16–S17. We find that the same nodes that drive the network to specific attractors under synchronous update are able to do so under asynchronous update as well.

## Discussion

In this paper, we generalize approaches typically considered in the study of spreading processes to Boolean dynamics.

First, we develop an individual-based mean-field approximation (IBMFA) for Boolean network dynamics. The approximation

neglects dynamical correlations between Boolean variables, but fully accounts for the topology and the dynamical rules of the network at hand. On sparse networks, the approximation allows to compute average trajectories in a time that grows linearly with the system size.

Second, we leverage the IBMFA to measure dynamical influence of nodes in Boolean networks. We measure influence of a set of seed nodes in terms of the entropy associated with long-term configurations that result from perturbing that set of seed nodes. Perturbations consist in pinning the Boolean state of the seed nodes to a given value during the dynamics of the system. All other nodes have an initial state that is maximally uncertain. High entropy values indicate that several configurations are possible; low entropy values indicate that only a few configurations are reachable; null entropy indicates that the seed set drives the dynamics toward one configuration only. We validate the use of this metric of influence on the *Drosophila melanogaster* segment polarity network (SPN). Further, we reproduce known anti-cancer effects of various drugs in combination with *Alpelisib* on the estrogen receptor breast cancer network.

Third, we deploy a greedy selection process to find minimal driver sets to reach specific attractors in Boolean networks. We validate the method by retrieving known attractors of the *Drosophila melanogaster* SPN and the yeast cell-cycle network, as verified by brute-force computation. We then use the method to

find minimal driver sets to control a network toward an unconstrained attractor in random Boolean networks (RBNs) and biochemical regulation and signaling networks from the Cell Collective repository. Although there are no guarantees of finding the largest attractor basin, the attractor found by our unconstrained greedy selection process is likely to have a large basin of attraction as it requires a minimal number of nodes to be perturbed and thus can be considered the easiest attractor to reach by random initial conditions and perturbations. Interestingly, we see no relation between the relative driver set size and the system size, indicating that control to an unspecified attractor (i.e., maximum influence) depends on the specific nature of the network dynamics. In the Cell Collective repository, we find that 65% of the networks can be controlled by less than 20% of their nodes, and that 80% can be controlled by less than 30% of their nodes. This is similar to previous estimates of driver set sizes in the more general problem of full attractor controllability[23]. The implications of our results are not as immediate and require further analysis. Via unconstrained optimization, we likely find driver sets toward attractors with a large basin of attraction. There are networks where attractors with a large basin are regarded as biologically meaningful[25,36]. However, there might be networks where those special attractors do not necessarily correspond to the most relevant forms of biological control.

Our greedy selection algorithm scales cubically with the system size and exponentially with the maximum degree of the network, making it applicable to medium-sized networks as long as they are sufficiently sparse.

Other scalable methods exist to identify the nodes that most influence nonlinear dynamics[16,40,47], but these methods do not tell, in general, how to control the dynamics toward a specific attractor. Unlike logical inference methods[16,40], the IBMFA does not guarantee to find specific causal pathways that determine dynamical behavior, but it does allow to estimate the state of every node in the network by averaging over all possible configurations allowed by pinning the seeds. This allows for a detailed description of the network's state from maximally uncertain initial conditions. This is also an advantage over structure-only methods which are scalable but may not predict well the dynamics of the network[17].

All methods developed in this paper can be immediately extended to deal with arbitrary updating schemes, e.g., deterministic asynchronous, stochastic asynchronous, and block deterministic updating schemes[41–43]. While fixed points are invariant to the choice of the updating scheme, the size of the basin of attraction of a fixed point is generally affected by the specific rules of the dynamics at hand[41,43]. This fact is apparent from the application of our methods too. For example, we find that unconstrained optimization leads to the identification of different fixed points for different updating schemes in the yeast *Saccharomyces cerevisiae* cell-cycle network. On the other hand, results of our methods applied to the *Drosophila melanogaster* SPN are almost identical for all the updating schemes we considered. A better understanding of the robustness of minimal driver sets against the change of updating rules requires further investigation.

The methods developed in this paper suffer from some limitations. The computational time of the unconstrained optimization algorithm grows cubically with the system size. This is certainly an improvement over several existing methods, yet it allows for the analysis of relatively small systems only. Further, constrained optimization requires prior knowledge of the targeted fixed points. Acquiring information on all attractors generally requires a time that grows exponentially with the system size, generating a clear limitation to the applicability of our method. However, we note that many biological networks have known attractors related to specific phenotypes and our method can be applied to these without having to formulate the entire attractor landscape, which is an advantage over methods that require this calculation (e.g.,[25,26]); as with these methods, attractors can be found via sampling for networks that are too large for exhaustive computation. Finally, as it is currently formulated, our method is useful for the study of fixed points only but not of limit cycles. The method can be generalized to the study of these more complicated attractors, but only via a non-trivial generalization of our currently proposed metric of dynamical influence.

In spite of the above limitations, there are some immediate extensions to this work that deserve future attention. For example, our methods can be easily adapted to networks with more than two states per node. Also, our definition of dynamical influence and our algorithm for the selection of optimal driver sets can be extended to study the effect of short-term perturbations, i.e., the state of the seeds is only initially set to a given value but can be altered by the dynamics of the network. Finally, our method may be used in a variety of applications to approximate node influence in Boolean networks that are too large to calculate exact solutions.

## Methods

**Boolean networks.** We consider a deterministic, multivariate, discrete-time dynamical system whose interactions are represented as a graph $\mathcal{G}$ composed of $N$ nodes. Full information about the network topology is contained in the $N \times N$ adjacency matrix $A$ whose generic element $A_{ij} = 1$ if a connection between node $i$ and node $j$ exists, while $A_{ij} = 0$ otherwise. Please note that the network is directed, in the sense that, in general, $A_{ij} \neq A_{ji}$. Self-loops are allowed. The network topology serves to specify dependencies among variables in the definition of the dynamical system. Specifically, we consider the case where, at the generic instant of time $t$, each node $i$ has associated a binary state variable $\sigma_i(t) = 0, 1$, and the value of the variable $\sigma_i(t)$ is fully determined by the value of the state variables of the neighbors of node $i$ at time $t - 1$. We can write

$$\sigma_i(t) = F_i\left(\vec{\sigma}_{\mathcal{N}_i}(t-1)\right), \qquad (1)$$

where $\vec{\sigma}_{\mathcal{N}_i}(t-1) = [\sigma_{j_1^{(i)}}(t-1), \ldots, \sigma_{j_{k_i}^{(i)}}(t-1)]$ is the vector representing the configuration of the system restricted to the neighborhood $\mathcal{N}_i = \{j_1^{(i)}, \ldots, j_{k_i}^{(i)}\} = \{j \in \mathcal{G} | A_{ji} = 1\}$ of node $i$, where $k_i$ is the in-degree, or simply the degree, of node $i$. $F_i(\cdot)$ is the binary-valued activation function of node $i$ and fully determines the rules of the dynamics of variable $\sigma_i$. Rules $F_i(\cdot)$s are static and do not evolve in time.

Given an initial condition $\vec{\sigma}(t=0) = [\sigma_1(t=0), \sigma_2(t=0), \ldots, \sigma_N(t=0)]$, the dynamics of the system consists in iterating the deterministic rules of Eq. (1). All results reported in the main paper are obtained under the synchronous updating scheme where all variables are updated in a synchronous manner. The above formalization, however, applies also to other updating schemes, e.g., deterministic asynchronous, stochastic asynchronous, and block deterministic updating schemes[41–43]. Depending on the functions $F_i(\cdot)$s and the initial condition $\vec{\sigma}(t=0)$, different long-term behaviors can be observed, including absorbing configurations and limit cycles. Fixed points of the network are insensitive to the specific updating scheme considered; however, the size of their basin of attraction is affected by it.

**Individual-based mean-field approximation.** Given a network and a set of Boolean functions describing the dynamics of the individual nodes, a brute-force analysis of the system would require considering all possible configurations and the effect of the functions $F_i(\cdot)$s on those configurations. This would allow to build a deterministic transition matrix between the $2^N$ possible configurations, thus providing a way to describe all possible trajectories of the system in terms of a state-transition graph. Clearly, such a brute-force approach does not scale properly with the system size, and thus it is not very useful for systematic analyses. We propose here a way to approximate system dynamics by assuming that expectation values of the various dynamical variables are uncoupled. Specifically, we consider the probability $s_i(t) = P(\sigma_i(t) = 1)$ and use Eq. (1) to write

$$s_i(t) = \sum_{\{n_j : j \in \mathcal{N}_i\}} \delta_{1, F_i(\vec{n}_{\mathcal{N}_i})} \prod_{j \in \mathcal{N}_i} [s_j(t-1)]^{n_j} [1 - s_j(t-1)]^{1-n_j}. \qquad (2)$$

Essentially, the probability $s_i(t) = P(\sigma_i(t) = 1)$ that node $i$ is found in the state $\sigma_i(t) = 1$ at time $t$ is given by a sum over all possible $2^{k_i}$ input configurations for the function $F_i(\cdot)$. Configurations are enumerated using $k_i$ binary variables $n_j$. Among those configurations, non-null contributions to the sum arise only when $F_i(\vec{n}_{\mathcal{N}_i}) = 1$. This fact is encoded by the term $\delta_{1, F_i(\vec{n}_{\mathcal{N}_i})}$, where we made use of the

Kronecker function defined as $\delta_{x,y} = 1$ if $x = y$ and $\delta_{x,y} = 0$ otherwise. We note that each configuration in the sum has a weight equal to a product of marginal probabilities, thus it is based on the approximation that the states of all nodes involved in the input configuration of the function $F_i(\cdot)$ are independent of each other. For example, if node $i$ has only three neighbors $j$, $k$, and $i$ itself, the hypothetical configuration $(n_i = 1, n_j = 0, n_k = 1)$ such that $F_i(1, 0, 1) = 1$ would correspond to a contribution equal to $s_i(t-1)\,[1 - s_j(t-1)]s_k(t-1)$ in the sum. Under the individual-based mean-field approximation (IBMFA), the probability of observing a configuration $\vec{\sigma}(t)$ given the values of the probabilities $\vec{s}(t)$ is

$$P\left(\vec{\sigma}(t)\,|\,\vec{s}(t)\right) = \prod_{i=1}^{N} [s_i(t)]^{\sigma_i(t)}\,[1 - s_i(t)]^{1 - \sigma_i(t)}. \tag{3}$$

We note that if we set $s_i(t) = \sigma_i(t) = 0, 1$ for all nodes $i$ in the network, then that is the only configuration with non-null probability in Eqs. (3), and (2) reduces to Eq. (1).

**Error metric for the individual-based mean-field approximation.** Error of the IBMFA is determined by comparing the approximation to $R$ simulations of network dynamics, each started from a random initial configuration obeying the probability distribution of Eq. (3). Specifically, we first estimate the average value

$$\overline{\sigma}_i(t) = \frac{1}{R} \sum_{r=1}^{R} \sigma_i^{(r)}(t), \tag{4}$$

where $\sigma_i^{(r)}(t)$ indicates the state of node $i$ at time $t$ in the $r$-th simulation. We then evaluate the mean squared error of the prediction as

$$e(t) = \frac{1}{N} \sum_{i=1}^{N} \left[ s_i(t) - \overline{\sigma}_i(t) \right]^2, \tag{5}$$

with $s_i(t)$ solution of the Eq. (2). The baseline value for the mean squared error of the IBMFA prediction is given by

$$b(t) = \frac{1}{R\,N} \sum_{i=1}^{N} \sum_{r=1}^{R} \left[ \sigma_i^{(r)}(t) - \frac{R\overline{\sigma}_i(t) - \sigma_i^{(r)}(t)}{R - 1} \right]^2. \tag{6}$$

Eq. (6) quantifies the variance of the sampled trajectories that are used to estimate the ground-truth average trajectory with Eq. (4).

**Entropy of network configurations.** We measure the uncertainty of a Boolean network as the normalized entropy of the probability distribution associated to its possible configurations, namely

$$H(\vec{s}) = \frac{1}{N} \sum_{i=1}^{N} h_2(s_i) \tag{7}$$

with $h_2(s)$ binary entropy function, i.e., $h_2(s) = -s \log_2(s) - (1 - s) \log_2(1 - s)$. Please note that Eq. (7) approximates the true entropy of the system from the above, as it assumes independence among the dynamical variables of the individual nodes. We note that $H(\vec{s}) \in [0, 1]$. Maximum entropy is reached for $s_i = 1/2$ for all $i$. Null entropy is measured for deterministic configurations $s_i = 0, 1$ for all $i$.

**Definition of seed set.** We define the set of seed nodes $\mathcal{X} = \{(x_1, \hat{\sigma}_{x_1}), (x_2, \hat{\sigma}_{x_2}), \ldots, (x_{|\mathcal{X}|}, \hat{\sigma}_{x_{|\mathcal{X}|}})\}$ as the set of nodes and their known, invariant, states, i.e., $\sigma_i(t) = s_i(t) = \hat{\sigma}_i = 0, 1$ for all $(i, \hat{\sigma}_i) \in \mathcal{X}$ and for all $t \geq 0$. Please note that we tacitly assumed that node $i$ may contribute at max one element to the set $\mathcal{X}$, as either $(i, \hat{\sigma}_i = 0)$ or $(i, \hat{\sigma}_i = 1)$.

The state of all nodes not belonging to the set $\mathcal{X}$ is uncertain, i.e., $0 \leq s_i(t) \leq 1$ for all $i \notin \mathcal{X}$. Unless noted otherwise, we focus our attention to the case of maximal uncertainty for the initial state of the non-pinned nodes, i.e., $s_i(0) = 1/2$ for all $i \notin \mathcal{X}$. The probability of starting from the configuration $\vec{\sigma}(0)$ given $\vec{s}(0)$ still obeys Eq. (3), with the additional constraint that $s_i(0) = \hat{\sigma}_i$ for $i \in \mathcal{X}$.

**Influence maximization.** We propose a greedy algorithm for the quasi-optimal selection of the smallest set of nodes that should be pinned in order to control the dynamics of a Boolean network toward zero entropy. As in standard greedy optimization techniques, our strategy consists in pinning one node at each stage of the algorithm; the selected seed is the best choice that can be made at that particular stage of the optimization algorithm.

Indicate with $\mathcal{X}_\nu = \{(b_1, \hat{\sigma}_{b_1}), (b_2, \hat{\sigma}_{b_2}), \ldots, (b_\nu, \hat{\sigma}_{b_\nu})\}$ the set of pinned nodes at the $\nu$-th stage of the algorithm. We initialize the algorithm at stage $\nu = 0$ with $\mathcal{X}_0 = \emptyset$. Then, we set $\nu = 1$ and follow the procedure:

1. Select the best seed $(b_\nu, \hat{\sigma}_{b_\nu})$ of stage $\nu$ of the algorithm according to

$$(b_\nu, \hat{\sigma}_{b_\nu}) = \arg \min_{(i, \hat{\sigma}_i) \notin \mathcal{X}_{\nu-1}} H(\vec{s}(T)\,|\,\mathcal{X}_{\nu-1} \cup (i, \hat{\sigma}_i)). \tag{8}$$

2. Add $(b_\nu, \hat{\sigma}_{b_\nu})$ to the set of pinned nodes $\mathcal{X}_{\nu-1}$, i.e., $\mathcal{X}_\nu = \mathcal{X}_{\nu-1} \cup (b_\nu, \hat{\sigma}_{b_\nu})$.
3. Increase $\nu \to \nu + 1$, and go back to point 1.

The above algorithm is iterated until there is a certain $\nu^*$ such that $H(\vec{s}(T)\,|\,\mathcal{X}_{\nu^*}) = 0$. In this case, the set of pinned nodes $\mathcal{X}_{\nu^*}$ is able to fully control the dynamics of the network toward a particular attractor. The set $\mathcal{X}_{\nu^*}$ is the optimal driver set according to our recipe. We note that the criterion of Eq. (8) prescribes the selection of the best seed as the one that, if added to the existing seed set, leads to the minimum resulting conditional entropy of the network. Conditional entropy is measured after $T$ dynamical stages of the dynamics. In particular, we approximate it via the solution of the IBMFA Eq. (2) where we impose $s_i(t = 0) = 1/2$ for all non-pinned nodes.

The resulting driver set is post-processed to eventually reduce its size. One node at a time is removed from the set as long as the resulting entropy is still equal to zero, i.e., the element $(i, \hat{\sigma}_i) \in \mathcal{X}$ can be removed from the set of drivers $\mathcal{X}$ only if $H(\vec{s}(T)\,|\,\mathcal{X} \setminus (i, \hat{\sigma}_i)) = 0$. The post-processing technique serves to improve suboptimal choices potentially made by the greedy optimization algorithm.

In order to select nodes to reach the attractor $\Theta = \{(1, \hat{\sigma}_1), (2, \hat{\sigma}_2), \ldots, (N, \hat{\sigma}_N)\}$, the same greedy selection procedure as above is used except that the set of candidate elements are only those compatible with $\Theta$, i.e., Eq. (8) is replaced by

$$(b_\nu, \hat{\sigma}_{b_\nu}) = \arg \min_{(i, \hat{\sigma}_i) \notin \mathcal{X}_{\nu-1}, (i, \hat{\sigma}_i) \in \Theta} H(\vec{s}(T)\,|\,\mathcal{X}_{\nu-1} \cup (i, \hat{\sigma}_i)). \tag{9}$$

A post-processing technique to reduce the size of the driver set is used in a similar manner as described for the unconstrained greedy optimization algorithm.

The computational time of the algorithm scales cubically with the system size, i.e., $O(N^3)$ (see Fig. S18). This fact is understood as follows. At stage $\nu$ of the algorithm, one needs to evaluate the entropy $H(\vec{s}(T)\,|\,\mathcal{X}_{\nu-1} \cup (i, \hat{\sigma}_i))$ appearing on the rhs of Eq. (8) for every of the $N - \nu$ nodes $i$ in the network that are not yet part of the seed set $\mathcal{X}_\nu$. Evaluating $H(\vec{s}(T)\,|\,\mathcal{X}_{\nu-1} \cup (i, \hat{\sigma}_i))$ requires a time that grows as $N$ as one needs to iterate the $N$ IBMFA Eq. (2). To find a driver set, the algorithm is iterated $\nu^*$ times, where $\nu^*$ grows as $N$ given that the size of a driver set is generally proportional to the system size.

**Networks analyzed in the paper**

*Random Boolean networks.* Random Boolean networks (RBNs) are extensively studied networks with well-known theoretical properties. RBNs are special cases of Boolean networks where the connections between nodes and the transfer functions governing node update are random. We consider RBNs under synchronous update, as in traditional literature[11,39,48].

In our model, the network has $N$ nodes and every node has exactly $k$ neighbors; we further ensure that there are no isolated nodes. For $k = 1$, we use a directed ring structure; for $k = 2$, we use an undirected ring structure; for $k \geq 3$, we generate random connections between the nodes. Activation functions are generated by assigning a random output value, either 0 or 1 with equal probability, to the function $F_i$ of Eq. (1) for all nodes $i$ irrespective of the $2^k$ possible arguments of the function.

*Gene regulatory networks.* Boolean and multi-state networks have been used to successfully model biological processes such as cell-fate determination, cell-cycle regulation, and cancer development[7,35,36,49]. Such gene regulatory networks (GRNs) characterize relevant components of a cell (e.g., a protein or gene) as nodes which are connected if one component has a regulatory (activating or inhibitory) effect on another. Boolean node states describe whether there is activity of that component above or below a relevant threshold. The attractors of the network mimic actual stable states of biological interest, such as wildtype or mutant phenotypes. These models are particularly useful in cases where kinetic parameters of biological components have not been established or where phenotypes of interest can be recovered without such parameters[50].

We utilize some of such models in this work. The first describes the body segmentation of *Drosophila melanogaster*[34]. The single-cell segment polarity network (SPN) consists of $N = 17$ nodes, three of which are external signals to the cell which have no inputs themselves. This network has been well studied and all of its 10 attractors are known. The second model describes signal transduction in estrogen receptor (ER+) breast cancer[35]. This network consists of $N = 80$ nodes, and its attractor landscape is too large to be fully described; however, the network contains several pathways of biological interest that can be manipulated by 7 external drug nodes. The activation of these nodes suggest that the drug is present, while their absence suggests that the drug is absent.

We also explore the *Drosophila melanogaster* parasegment network ($N = 60$), which is the equivalent of four interconnected single-cell SPN models where each cell has 15 nodes, some of which are dependent on neighboring cells and one which is an external signal[34]. Although the complete attractor landscape of this network is largely unexplored, several biologically relevant attractors are known, representing the wildtype, wildtype variant, ectopic, ectopic variant, broad stripes, and no segmentation phenotypes. In addition, we analyze the T cell large granular lymphocyte (T-LGL) leukemia network ($N = 60$), which describes T cell survival signaling in leukemia[37], and the yeast *Saccharomyces cerevisiae* cell-cycle network ($N = 12$), which describes cell-cycle regulation in budding yeast[36]. Finally, we utilize the Cell Collective repository, an open source collection of biological signaling and regulatory networks[38], as accessed on August 5th, 2020.

All results of the main paper have been obtained assuming that the system evolves under synchronous updates. In the SI, we study the dynamical properties of the *Drosophila melanogaster* SPN and the yeast *Saccharomyces cerevisiae* cell-cycle network considering deterministic asynchronous, stochastic asynchronous, and block deterministic updating schemes[41–43].

**Reporting summary**. Further information on research design is available in the Nature Research Reporting Summary linked to this article.

## Data availability

Network data considered in this paper have been downloaded from the publicly accessible repositories https://github.com/rionbr/CANA/tree/master/cana/datasets and https://cellcollective.org.

## Code availability

The code developed for this paper is made available at https://doi.org/10.5281/zenodo.6581810[51].

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

## Acknowledgements

This material is based upon work supported by the Air Force Office of Scientific Research under award number FA9550-21-1-0446 (T.P. and F.R.) and by the National Institutes of Health, National Library of Medicine Program, grant 01LM011945-01 (L.M.R.).The funders had no role in study design, data collection and analysis, decision to publish, or any opinions, findings, and conclusions or recommendations expressed in the manuscript.

## Author contributions

T.P., L.M.R., and F.R. conceived and designed the experiments. T.P. performed the experiments. T.P., L.M.R., and F.R. wrote the paper.

## Competing interests

The authors declare no competing interests.
