## [Peer Review File · Nature Communications]

REVIEWER COMMENTS

Reviewer #1 (Remarks to the Author):

The paper presents an interesting method for studying, computationally, the identification of minimal sets of nodes in a Boolean network (BN), able to drive the dynamics towards desired long-term behaviors. Computer simulations were run considering random Boolean networks as well as gene regulatory network (GRN) models.

In the context of GRN, the updating scheme of the BN is important. Moreover, the synchronous or parallel updating scheme is the less biologically plausible scheme. Although fixed points are invariant against the updating scheme, their basin of attraction may change when changing the updating scheme from one to another. Suppose for a biological system, the attractor of interest is a limit cycle, like in the model of the mammalian cell-cycle network (<https://doi.org/10.1093/bioinformatics/btl210>). In that case, the updating scheme has a significant impact since limit cycles tend to disappear when you change the updating scheme.

Therefore an important limitation of the submitted paper is that all the simulations were done considering only the synchronous updating scheme. It would be interesting to see some results when considering other updating schemes, at least deterministic ones, like sequential or block sequential updating schemes. Moreover, for the deterministic case, there are methods that allow exploring all the possible deterministic updates, for example, the equivalent classes approach: <https://doi.org/10.1016/j.dam.2010.10.010>

Also, it would be interesting if the authors could compare their results for the yeast cell cycle network that has been completely studied for all the possible deterministic updating schemes in: <https://doi.org/10.1007/s11538-012-9794-1>

In particular, in that work, the authors introduce the notion of alliances, a subconfiguration of binary values associated with a subset of genes that remain fixed for any binary value of the other genes. I am wondering if the alliances identified in the above paper for the yeast network (Yeast2 in the above paper) are related to what the submitted paper says: "the smallest set of nodes that should be pinned in order to control the dynamics of a Boolean network towards zero entropy "

In summary, I think the work would be benefited by presenting some results using other updating schemes, as well as analyzing their results of the yeast cell cycle network with the theoretical results obtained in <https://doi.org/10.1007/s11538-012-9794-1>

Reviewer #2 (Remarks to the Author):

In this manuscript, the authors describe a mean-field analysis of dynamics in given Boolean networks, allowing them to efficiently estimate the effects of control over individual nodes. Particularly, they focus on finding a control set for each network whose pinning results in single attractor. Using a combination of the mean-field estimate of the entropy under control pinning, they find estimates of minimal control sets by forcing this estimated entropy to zero. They also find that the typical size of control sets is relatively small, around 20-40% of the total number of nodes, in both random networks and a set of known biological networks.

As a possibly highly efficient approximation, the method developed here would be a welcome addition to a growing literature on nonlinear methods for finding control strategies in discrete network dynamics. However, I find a number of issues in the communication of the results and their relation to prior known results that limit my confidence in the conclusions reached.

First, I did not find the validation of the method completely convincing. The authors checked their results in three ways: (1) by computing the error between true trajectory distributions and those produced by the mean-field method (Fig. 1), (2) by comparing the approximated entropies to those of the true distributions (Fig. S1), and (3) by comparing to sizes of control sets known from previous studies. (1) showed that the approximate trajectories are relatively close to their true values, but it is hard to know whether this is enough to correctly identify control nodes (e.g. it's possible that the errors are due to many nearby attractors that would not be removed with the estimated control). (2) showed a correlation between the true and approximated entropies after 10 steps (Fig. S1d), but the absolute values are not particularly close. Also, it is again not clear that getting the entropies nearly right will correspond to getting the control sets nearly right. (3) Here, the authors apply the method to two known biological networks (the "Drosophila" and "yeast cell cycle" networks) and compute control sets, but only roughly compare the size of control sets with previous results. In these relatively small networks, a more convincing check could be made by comparing to the true minimal control set sizes explicitly for all individual attractors (or perhaps the attractor with largest basin). See e.g.:

Kim, Junil, Sang-Min Park, and Kwang-Hyun Cho. "Discovery of a Kernel for Controlling Biomolecular Regulatory Networks." *Scientific Reports* 3 (2013): 2223. <https://doi.org/10.1038/srep02223>.

Second, there exist other methods for computing minimal control sets exactly that could be used to more strongly validate these results. Particularly for small networks (those of size ~ 10) it is not difficult to compute minimal control sets directly from the state transition graph. There are also other methods

that use the structure of the interaction network to compute exact minimal control set sizes for given attractors in some networks up to size ~ 50 (and for some of the same biological networks studied here). See:

Borriello, Enrico, and Bryan C. Daniels. "The Basis of Easy Controllability in Boolean Networks." *Nature Communications*, no. 12 (2021): 5227. <https://doi.org/https://doi.org/10.1038/s41467-021-25533-3>.

Zañudo, Jorge G.T., and Réka Albert. "Cell Fate Reprogramming by Control of Intracellular Network Dynamics." *PLoS Computational Biology* 11, no. 4 (2015): 1–24. <https://doi.org/10.1371/journal.pcbi.1004193>.

Other issues:

Aiming for control that produces one particular attractor versus producing *any* single attractor can give very different results. As the authors point out, leaving the attractor unspecified will likely lead to one that has a large basin. This is fine, but it likely affects the result shown in Fig. S6: Biased networks will be easier to control toward biased attractors, but they may be harder to control toward other attractors. It is also unclear that the most relevant forms of biological control would be those leading toward attractors with large basins.

The efficiency of the method is important as it is a distinguishing feature from other methods. The authors claim that the running time scales as N^3 (Fig. S10), but the cubic fit is not particularly convincing in the biological cases, and I could not find an argument for why this should be cubic. (Section II.A claims that "the approximation grows linearly with N and exponentially with the degree of the nodes", which by context I think means "the running time grows linearly..."---but there is no mention of N^3 .)

The use of the "baseline" values defined in Eq. 6 and shown in Fig. 1 was not clear. How does this show that "IBMFA is more informative than observing the outcome of a single instance of the network dynamics", and why is this an important comparison?

Minor issues/typos:

Fig. 1 legend--reference to Eq.(1) should be Eq.(3)?

Fig. 2b legend--too many curly brackets?

Reviewer #3 (Remarks to the Author):

In this study, the authors develop a method for finding a set of nodes which have the maximum influence on the boolean dynamics in long term. While also introduce how to quantify the influence of the sets by entropy.

Overall, the paper introduces a new and interesting approximation method that advances the applications of boolean networks' modeling for biological systems. The approach is rather simple, but brings a set of new results, potentially of interest for the broad readership of the Nature Communications. I think the manuscript has substantial merits to be accepted for publication in the Nature Communications, but after a major revision.

Strengths:

- I found the application of this method on empirical boolean networks strong.
- The approach is innovative, and it could boost new and impactful research in the area.

Weaknesses:

- The method is only tested on a specific sort of RBNs (Sec. G, 1). What about having different rules?
- The paper is generally not well-structured. Directly after introduction the results come and one needs to go the last section for the method, formulas and details ...

Comments and questions:

- the title and abstract are not reflecting well the idea of the paper
- having some schematic figures with some examples can help a lot to understand the idea easier, either in introduction or method section.
- discussing about some limit cases would help a lot. For instance, when dynamics have only one attractor or when number of attractors are too high but they are small, or when you choose some specific x set (topologically)...

- are x sets chosen from specific sub-graphs in the network? or have you checked if the optimised sets have any specific topological feature or position in the network?
- how good does this method work in different (ordered-disordered) regimes (see this paper for instance: PRL 107 (18), 188701, 2011) one can argue by choice of boolean rules and how perturbing the x sets ...
- I missed the point how the desired attractor is chosen.
- the algorithm, e.g. flowchart can be included in the supplementary material.
- in Fig 3. b, do all the 10 attractors have size 1?

First referee

The paper presents an interesting method for studying, computationally, the identification of minimal sets of nodes in a Boolean network (BN), able to drive the dynamics towards desired long-term behaviors. Computer simulations were run considering random Boolean networks as well as gene regulatory network (GRN) models.

We thank the reviewer for the time dedicated to our manuscript, and we appreciate the very positive report. We took advantage of the useful comments made by the reviewer to significantly expand our analysis and improve the overall quality of our work.

In the context of GRN, the updating scheme of the BN is important. Moreover, the synchronous or parallel updating scheme is the less biologically plausible scheme. Although fixed points are invariant against the updating scheme, their basin of attraction may change when changing the updating scheme from one to another. Suppose for a biological system, the attractor of interest is a limit cycle, like in the model of the mammalian cell-cycle network (<https://doi.org/10.1093/bioinformatics/bt1210>). In that case, the updating scheme has a significant impact since limit cycles tend to disappear when you change the updating scheme.

We agree with the comment made by the reviewer. The behavior of a Boolean network may be radically different depending on the updating scheme regulating system's dynamics.

Therefore an important limitation of the submitted paper is that all the simulations were done considering only the synchronous updating scheme. It would be interesting to see some results when considering other updating schemes, at least deterministic ones, like sequential or block sequential updating schemes. Moreover, for the deterministic case, there are methods that allow exploring all the possible deterministic updates, for example, the equivalent classes approach: <https://doi.org/10.1016/j.dam.2010.10.010>

We agree with the reviewer that considering only the synchronous updating scheme was one important limitation of our original analysis. We included results from different updating schemes in the revised version of the paper, see Sec. II A-C and Supplementary Information (SI).

Also, it would be interesting if the authors could compare their results for the yeast cell cycle network that has been completely studied for all the possible deterministic updating schemes in: <https://doi.org/10.1007/s11538-012-9794-1>

We included an expanded analysis of the yeast cell-cycle network in the revised version of the manuscript (see Sec. II A-C and SI).

In particular, in that work, the authors introduce the notion of alliances, a subconfiguration of binary values associated with a subset of genes that remain fixed for any binary value of the other genes. I am wondering if the alliances identified in the above paper for the yeast network (Yeast2 in the above paper) are related to what the submitted paper says: “the smallest set of nodes that should be pinned in order to control the dynamics of a Boolean network towards zero entropy”

By definition, the state of the nodes within an alliance is fixed. However, the state of the other nodes outside the alliance is not fixed. This ambiguity in the state of the nodes outside an alliance prevents the system from reaching zero entropy, which is our condition to determine a driver set. As a consequence, a driver set and an alliance may overlap but only partially, except for the trivial case of a fixed point.

In summary, I think the work would be benefited by presenting some results using other updating schemes, as well as analyzing their results of the yeast cell cycle network with the theoretical results obtained in <https://doi.org/10.1007/s11538-012-9794-1>

All methods presented in the paper, including the equations developed under the individual-based mean-field approximation (IBMFA), can be immediately adapted to deal with arbitrary updating schemes. We were therefore able to expand the analysis as suggested by the reviewer. In the revised version only results concerning synchronous dynamics are shown in the main paper. However, we included all novel results about different updating schemes in the revised SI. These results are mentioned in the main paper (see Sec. II A-C). Also, we included references to all papers listed by the reviewer. Our stylistic choice was made for simplicity of presentation. We wanted to emphasize the novel methods without making the paper too long. Discussing the different behaviors that a network may exhibit depending on the updating scheme of its dynamics, while certainly interesting, is not the main focus of the current paper.

To facilitate the assessment by the reviewer, we report the results of our novel analysis below, see Figures R1- R6. All these figures have been included in the revised version of the SI. We basically generalized the analysis already made for the synchronous updating scheme to other updating schemes. In particular:

- We focused our attention on two main networks: (1) the *Drosophila melanogaster* segment polarity network (SPN) [i.e., Fig. R1, R2, R4, and R6] and (2) the yeast *Saccharomyces cerevisiae* cell-cycle network [i.e., Fig. R1, R3, R5, and R6].
- We considered three asynchronous updating schemes: (1) deterministic (also called sequential), (2) stochastic with replacement, and (3) stochastic without replacement. In all cases, the state of a single node is updated while the state of all other nodes is kept invariant, and one unit of time corresponds to a number of updates equal to the system size. The node whose state is updated is selected in different ways depending on the updating scheme: in (1), the node is selected according to a predetermined sequence; in (2), the node is selected at random among all the possible nodes; in (3), the node is selected according to a sequence,

but such a sequence is randomized at every time step. Also, for the yeast cell-cycle network, we considered (4) the block-sequential update proposed in Ref. [1], where the nodes' states are updated in blocks following a predetermined sequence.

- We studied how the updating scheme affects (1) the accuracy of IBMFA [i.e., Fig. R1], (2) the dynamical influence of seed sets [i.e., Figs. R2 and R3], (3) the robustness of optimal driver sets towards known fixed points [i.e., Figs. R4 and R5], (4) the robustness of the minimal driver set [i.e., Fig. R6].

In general, we found that for the *Drosophila melanogaster* SPN the updating scheme affects only the short-term behavior of system, while long-term properties are largely not affected. The long-term dynamical features of the yeast *Saccharomyces cerevisiae* cell-cycle network appear to be sensitive to the specific updating scheme at hand. This latter observation is in line with the findings of Ref. [1] that the attractor basins change size depending on the updating scheme, as pointed out also by the reviewer. Interestingly, the error of the mean-field approximation (R1) is largest for the synchronous update scheme we describe in the main text. Therefore, the good results we describe in the main paper refer to the most difficult updating scheme for (mean-field) prediction.

Figure R1: Accuracy of the individual-based mean-field approximation across various update schedules. (a) We evaluate the error $e(t)$, as defined in Eq. (5) of the main text, committed by the individual-based mean-field approximation (IBMFA) in predicting the ground-truth configuration of the system at stage t of the dynamics. Each curve corresponds to a different updating schedule in the *Drosophila melanogaster* segment polarity network (SPN): synchronous (syn), deterministic asynchronous (det asyn), stochastic asynchronous without replacement (sto w/o rep) and stochastic with replacement (sto w rep). In all asynchronous updating schemes, the state of a single node is updated while the state of all other nodes is kept invariant, and one unit of time corresponds to a number of updates equal to the system size. The node whose state is updated is selected in different ways depending on the updating scheme: in (det asyn), the node is selected according to a predetermined sequence; in (sto w rep), the node is selected at random among all the possible nodes; in (sto w/o rep), the node is selected according to a sequence, but such a sequence is randomized at every time step. All results are averaged over $R = 100$ independent simulations of $M = 100$ randomly selected updating schemes. Initial configurations are sampled from Eq. (3) of the main paper, where the probability of node i to be active equals $s_i(t = 0) = 1/2$ for all $i = 1, \dots, N$. (b) Same as in panel a, but for the yeast cell-cycle network. In addition to the other update schedules, the block-sequential update (block-seq) from Ref. [1] is included. In this case, nodes are updated sequentially in blocks, where all nodes in the same block are updated synchronously. The blocks updated in order are: ['CellSize', 'Swi5', 'Cdc20/14', 'Clb5,6'], ['MBF', 'Sic1'], ['Cln3', 'SBF', 'Clb1,2', 'Mcm1/SFF'], ['Cln1,2'], and ['Cdh1']. One unit of time is given by a full update of all blocks.

Figure R2: **Dynamical influence in the *Drosophila Melanogaster* SPN under different update schedules.** (a) We monitor the residual entropy of the *Drosophila melanogaster* segment polarity network (SPN) as a function of time. Entropy is measured in bits and is normalized by the size of the network. We report results for an empty seed set, i.e., for $\mathcal{X} = \emptyset$. Different curves correspond to different choices of the update schedule: synchronous (syn), deterministic asynchronous (det asyn), stochastic asynchronous without replacement (sto w/o rep) and stochastic with replacement (sto w rep). For definitions of the updating schemes, see caption of Figure R1. All results are averaged over $R = 100$ independent simulations of $M = 100$ models. (b) Same as in panel a, but for $\mathcal{X} = \{(en, \hat{\sigma}_{en} = 1)\}$. (c) Same as in panel a, but for $\mathcal{X} = \{(CIR, \hat{\sigma}_{CIR} = 0)\}$. (d) Same as in panel a, but for $\mathcal{X} = \{(CIR, \hat{\sigma}_{CIR} = 1)\}$.

Figure R3: Dynamical influence in the yeast cell-cycle network under different update schedules. (a) We monitor the residual entropy of the yeast cell-cycle network as a function of time. Entropy is measured in bits and is normalized by the size of the network. We report results for $\mathcal{X} = \emptyset$. Different curves correspond to different choices for the update schedule: synchronous (syn), deterministic asynchronous (det asyn), stochastic asynchronous without replacement (sto w/o rep), stochastic with replacement (sto w rep), and block-sequential (block-seq). For definitions of the updating schemes, see caption of Figure R1. All results are averaged over $R = 100$ independent simulations of $M = 100$ models. (b) Same as in panel a, but for $\mathcal{X} = \{(\text{CellSize}, \hat{\sigma}_{\text{CellSize}} = 1)\}$. (c) Same as in panel a, but for $\mathcal{X} = \{(\text{Cln3}, \hat{\sigma}_{\text{Cln3}} = 0)\}$. (d) Same as in panel a, but for $\mathcal{X} = \{(\text{Cln3}, \hat{\sigma}_{\text{Cln3}} = 1)\}$.

Figure R4: Driving the *Drosophila Melanogaster* SPN to a desired attractor under different update schedules. (a) We monitor the residual entropy of the *Drosophila Melanogaster* segment polarity network (SPN) as a function of time. Entropy is measured in bits and is normalized by the size of the network. We report results for the driver set solution for attractor 1 found by our greedy approximation algorithm under synchronous updating. Different curves correspond to different choices for the update schedule: synchronous (syn), deterministic asynchronous (det asyn), stochastic asynchronous without replacement (sto w/o rep) and stochastic with replacement (sto w rep). All results are averaged over $R = 100$ independent simulations of $M = 100$ models. (b-j) Same as in panel a, but for attractors 2-10. The attractors are displayed in the same order as Fig. 3 in the main text.

Figure R5: **Driving the yeast *Saccharomyces cerevisiae* cell-cycle network to a desired attractor under different updating schedules.** (a) We monitor the residual entropy of the yeast cell-cycle network as a function of time. Entropy is measured in bits and is normalized by the size of the network. We report results for the driver set solution for the attractor found by our unconstrained greedy approximation algorithm under synchronous updating. Different curves correspond to different choices for the update schedule: synchronous (syn), deterministic asynchronous (det asyn), stochastic asynchronous without replacement (sto w/o rep), stochastic with replacement (sto w rep), and block-sequential (block-seq). (b-l) Same as in panel a, but for the original 11 attractors of the yeast cell-cycle network. The attractors are displayed in the same order as in Fig. S5.

Figure R6: Attractor found by the greedy algorithm under different updating schemes. (a) We apply our unconstrained optimization algorithm for finding the minimal driver set toward an unspecified fixed-point of the *Drosophila melanogaster* segment polarity network (SPN). We consider different choices for the update schedule: synchronous (syn), deterministic asynchronous (det asyn), stochastic asynchronous without replacement (sto w/o rep) and stochastic with replacement (sto w rep). We measure the frequency of observation of attractor 4 (blue) for $M = 100$ models of the various updating schemes. Different attractors, each identified by a different color, are reached only for a few realizations of the asynchronous updating rules. (b) Same analysis as in panel a, but for the yeast *Saccharomyces cerevisiae* cell-cycle network. In this case, several different attractors, each identified by a different color, are reached depending on the updating scheme. We included in the analysis also the block-sequential (block-seq) updating scheme.

Second referee

In this manuscript, the authors describe a mean-field analysis of dynamics in given Boolean networks, allowing them to efficiently estimate the effects of control over individual nodes. Particularly, they focus on finding a control set for each network whose pinning results in single attractor. Using a combination of the mean-field estimate of the entropy under control pinning, they find estimates of minimal control sets by forcing this estimated entropy to zero. They also find that the typical size of control sets is relatively small, around 20-40% of the total number of nodes, in both random networks and a set of known biological networks.

As a possibly highly efficient approximation, the method developed here would be a welcome addition to a growing literature on nonlinear methods for finding control strategies in discrete network dynamics. However, I find a number of issues in the communication of the results and their relation to prior known results that limit my confidence in the conclusions reached.

We thank the reviewer for the time dedicated to our manuscript. We appreciate the positive opinion about the potential of our work and the constructive nature of the criticisms raised. We took advantage of these criticisms to extend/generalize our analysis. We believe that the revised version of the paper represents a substantial improvement over the manuscript we originally submitted, and we hope that the reviewer will recommend it for publication.

First, I did not find the validation of the method completely convincing. The authors checked their results in three ways: (1) by computing the error between true trajectory distributions and those produced by the mean-field method (Fig. 1), (2) by comparing the approximated entropies to those of the true distributions (Fig. S1), and (3) by comparing to sizes of control sets known from previous studies. (1) showed that the approximate trajectories are relatively close to their true values, but it is hard to know whether this is enough to correctly identify control nodes (e.g. it's possible that the errors are due to many nearby attractors that would not be removed with the estimated control). (2) showed a correlation between the true and approximated entropies after 10 steps (Fig. S1d), but the absolute values are not particularly close. Also, it is again not clear that getting the entropies nearly right will correspond to getting the control sets nearly right. (3) Here, the authors apply the method to two known biological networks (the "Drosophila" and "yeast cell cycle" networks) and compute control sets, but only roughly compare the size of control sets with previous results. In these relatively small networks, a more convincing check could be made by comparing to the true minimal control set sizes explicitly for all individual attractors (or perhaps the attractor with largest basin). See e.g.:

Kim, Junil, Sang-Min Park, and Kwang-Hyun Cho. "Discovery of a Kernel for Controlling Biomolecular Regulatory Networks." *Scientific Reports* 3 (2013): 2223. <https://doi.org/10.1038/srep02223>.

Thanks for making this excellent comment. We took advantage of it to perform additional validation of the methods we developed in our paper.

As indicated by the reviewer, we first validated our predictions for the minimal driver sets of the *Drosophila Melanogaster* SPN and the yeast cell-cycle network against the ground truth obtainable via brute-force approaches, see Figure R7. For the *Drosophila Melanogaster* SPN, our method for the identification of the minimal set of driver nodes necessary to reach a known fixed point retrieves the correct sets for 7 of the 10 fixed points, and it overestimates by one node the minimal set of drivers for the remaining 3 fixed points. We note that overestimating the size of the ground-truth driver set by a small margin is reasonable given the level of approximation used by our approach (the IBMFA neglects dynamical correlations; greedy optimization is necessarily sub-optimal). In the case of the yeast cell-cycle network our method correctly retrieves the minimal driver sets for 4 out of the 11 attractors; it overestimates the driver sets by one or two nodes for 5 fixed points; it underestimates the driver set by one node for 2 attractors. Comparisons between ground-truth driver sets and mean-field predictions are reported in Table R1. We recognize that underestimating the size of the driver sets is a not desirable outcome. On the other hand, we note that our underestimations still offer very good approximations of the ground truth, in the sense that the driver sets found by our approach lead to the attractors in 93% and 97% of the initial configurations, respectively. In other words, the additional node that is required to always reach the fixed points is used only in respectively 7% and 3% of the total possible initial configurations. We included comments about these validation tests in the revised version of the paper. Figure R7 and Table R1 have been added to revised version of the SI.

Figure R7: **Accuracy of the greedy selection process to predict ground-truth minimal driver sets.** (a) We use a brute-force approach on the state transition graph to determine the ground-truth size of the minimal driver set for each of the 10 attractors of the *Drosophila melanogaster* segment polarity network (SPN) (orange squares). Minimal driver sets to the same attractors are approximated using our constrained greedy optimization method (blue circles). (b) Same analysis as in panel (a), but for the 11 fixed points of the yeast *Saccharomyces cerevisiae* cell-cycle network.

Attractor	Method	Driver set
1	BF	{CellSize : 0; SBF : 0; MBF : 0; Sic1 : 1; Cdh1 : 1}
	GR	{CellSize : 0; SBF : 0; MBF : 0; Sic1 : 1; Cdh1 : 1}
2	BF	{CellSize : 1; SBF : 1; MBF : 1; Sic1 : 0; Clb5,6 : 0; Clb1,2 : 1}
	GR	{CellSize : 1; SBF : 1; MBF : 1; Sic1 : 0; Clb5,6 : 0; Clb1,2 : 1}
3	BF	{CellSize : 1; SBF : 1; MBF : 1; Clb5,6 : 1; Clb1,2 : 1}
	GR	{CellSize : 1; SBF : 1; MBF : 1; Clb5,6 : 1}
4	BF	{CellSize : 0; SBF : 0; Sic1 : 0; Cdh1 : 0; Mcm1/SFF : 0}
	GR	{CellSize : 0; SBF : 0; MBF : 0; Sic1 : 0; Clb5,6 : 0; Cdh1 : 0; Clb1,2 : 0}
5	BF	{CellSize : 1; SBF : 1; MBF : 0; Sic1 : 0; Clb1,2 : 1}
	GR	{CellSize : 1; SBF : 1; MBF : 0; Sic1 : 0; Clb1,2 : 1}
6	BF	{CellSize : 0; SBF : 1; Mcm1/SFF : 0}
	GR	{CellSize : 0; SBF : 1; MBF : 0; Clb5,6 : 0; Clb1,2 : 0}
7	BF	{CellSize : 0; SBF : 0; MBF : 1; Sic1 : 1; Cdh1 : 1}
	GR	{CellSize : 0; SBF : 0; MBF : 1; Sic1 : 1; Clb5,6 : 0; Cdh1 : 1}
8	BF	{CellSize : 1; SBF : 0; MBF : 1; Sic1 : 0; Clb5,6 : 1; Clb1,2 : 1}
	GR	{CellSize : 1; SBF : 0; MBF : 1; Sic1 : 0; Clb5,6 : 1}
9	BF	{CellSize : 0; SBF : 0; MBF : 1; Sic1 : 1; Cdh1 : 0}
	GR	{CellSize : 0; SBF : 0; MBF : 1; Sic1 : 1; Clb5,6 : 0; Cdh1 : 0}
10	BF	{CellSize : 0; SBF : 0; MBF : 0; Sic1 : 0; Cdh1 : 1}
	GR	{CellSize : 0; SBF : 0; MBF : 0; Sic1 : 0; Clb5,6 : 0; Cdh1 : 1}
11	BF	{CellSize : 0; SBF : 0; MBF : 0; Sic1 : 1; Cdh1 : 0}
	GR	{CellSize : 0; SBF : 0; MBF : 0; Sic1 : 1; Cdh1 : 0}

Table R1: Driver sets calculated by brute-force computation (BF) compared to our greedy selection algorithm (GR) for the yeast cell-cycle network. The attractors are in the same order as in Fig. R7. For each attractor, we specify the label of the driver nodes and their states. For sake of compactness, we use a different notation with respect to the rest of the paper. For example, the notation {CellSize : 0; SBF : 0; MBF : 0; Sic1 : 1; Cdh1 : 1} corresponds to the set $\{(CellSize, \hat{\sigma}_{CellSize} = 0), (SBF, \hat{\sigma}_{SBF} = 0), (MBF, \hat{\sigma}_{MBF} = 0), (Sic1, \hat{\sigma}_{Sic1} = 1), (Cdh1, \hat{\sigma}_{Cdh1} = 1)\}$. Attractor 1 corresponds to the G1 phase of the cell cycle and is the attractor with the largest basin.

Second, there exist other methods for computing minimal control sets exactly that could be used to more strongly validate these results. Particularly for small networks (those of size ≤ 10) it is not difficult to compute minimal control sets directly from the state transition graph. There are also other methods that use the structure of the interaction network to compute exact minimal control set sizes for given attractors in some networks up to size ≤ 50 (and for some of the same biological networks studied here). See:

Borriello, Enrico, and Bryan C. Daniels. "The Basis of Easy Controllability in Boolean Networks." *Nature Communications*, no. 12 (2021): 5227. <https://doi.org/https://doi.org/10.1038/s41467-021-25533-3>.

Zañudo, Jorge G.T., and Réka Albert. "Cell Fate Reprogramming by Control of Intracellular Network Dynamics." *PLoS Computational Biology* 11, no. 4 (2015): 1–24. <https://doi.org/10.1371/journal.pcbi.1004193>.

Once more, we thank the reviewer for suggesting to us to perform additional validation tests. We followed the advice and performed systematic comparisons between the driver sets obtained by our method and those obtained with the method proposed by Borriello and Daniels (BD). We were able to compare our predictions directly with the predictions of the BD method for the networks within the Cell Collective repository. Specifically, we looked at results of the BD method regarding the "control kernel," which approximates the minimal driver set of a network towards a desired attractor. The BD method allows to efficiently estimate the average size of control kernels in a network. Such an average is estimated over all attractors if the network is small enough; if the network is too large instead, the average value of the control kernel is estimated on a subsample of the network's attractors.

In the comparison with the BD method, we used our method in two different ways. In one case, we use it to find the optimal driver set for each fixed point of a network, and then estimate the average size of these sets. For large networks, we approximate the average size of the driver set on the same subsample of attractors used by the BD method to estimate the mean control kernel. In the other case, we apply the unconstrained version of our method to find the smallest among the driver sets. Results of our analysis are reported in the revised version of Figure 4 of the main paper. For simplicity, we report those results also in this letter, see Figure R8.

Overall, we find excellent agreement between results of our method and those by the BD method. The average optimal size of our driver sets matches well the size of the mean control kernel. The minimum-size driver set obtained with our method via unconstrained optimization is consistently smaller than the average value of the minimal driver sets over all the fixed points.

Figure R8: We identify the optimal sets of drivers by implementing our greedy strategy on 43 networks from the Cell Collective that were analyzed by [2] and that have fixed point attractors. We compare our predictions with those obtained by Borriello and Daniels [2]. We plot the average size of the optimal driver sets predicted by our method vs. the average size of the control kernels (blue circles). A control kernel is an approximation of the minimal driver set towards a desired attractor [2]; optimal driver sets are instead estimated using our constrained optimization algorithm. Each point in the plot is a network. Average values are computed over all the possible fixed points of a network. The dashed grey line indicates an exact match between the two predictions. For the same set of networks, we visualize also the size of the minimal optimal driver set (computed using our unconstrained optimization algorithm) vs. the average size of the control kernels (orange squares).

Other issues:

Aiming for control that produces one particular attractor versus producing any single attractor can give very different results. As the authors point out, leaving the attractor unspecified will likely lead to one that has a large basin. This is fine, but it likely affects the result shown in Fig. S6: Biased networks will be easier to control toward biased attractors, but they may be harder to control toward other attractors. It is also unclear that the most relevant forms of biological control would be those leading toward attractors with large basins.

We fully agree with the reviewer. We added a few clarifications in the conclusions of the main text to address this comment.

The efficiency of the method is important as it is a distinguishing feature from other methods. The authors claim that the running time scales as N^3 (Fig. S10), but the cubic fit is not particularly convincing in the biological cases, and I could not find an argument for why this should be cubic. (Section II.A claims that "the approximation grows linearly with N and exponentially with the degree of the nodes", which by context I think means "the running time grows linearly..."—but there is no mention of N^3 .)

Thanks for comment. We addressed both issues identified by the reviewer. Specifically:

1. We regenerated Figure S18 to display data on a double-logarithmic plot.
2. We included in the Methods section an explicit explanation of why our proposed method has running time growing as N^3 .

The use of the "baseline" values defined in Eq. 6 and shown in Fig. 1 was not clear. How does this show that "IBMFA is more informative than observing the outcome of a single instance of the network dynamics", and why is this an important comparison?

We agree with the reviewer that our description of baseline was not clear. In the revised version of the main text, we rephrased the text (both in the Results and Methods sections of the main paper) to address this issue.

Eq. (6) is the variance of the distribution of the sample configurations used to estimate the average trajectory. This represents the natural term of comparison for the mean squared error associated to our IBMFA predictions.

Minor issues/typos:

Fig. 1 legend—reference to Eq.(1) should be Eq.(3)?

Corrected, thanks!

Fig. 2b legend—too many curly brackets?

Corrected, thanks!

Third referee

In this study, the authors develop a method for finding a set of nodes which have the maximum influence on the boolean dynamics in long term. While also introduce how to quantify the influence of the sets by entropy.

Overall, the paper introduces a new and interesting approximation method that advances the applications of boolean networks' modeling for biological systems. The approach is rather simple, but brings a set of new results, potentially of interest for the broad readership of the Nature Communications. I think the manuscript has substantial merits to be accepted for publication in the Nature Communications, but after a major revision.

We are grateful to the reviewer for the time dedicated to our manuscript, and for the very positive report. Below, we provide a point-to-point reply to all issues raised by the reviewer. We addressed them in the revised version of our paper. We hope that the reviewer will support the publication of the manuscript in the journal.

Strengths:

- I found the application of this method on empirical boolean networks strong.

- The approach is innovative, and it could boost new and impactful research in the area.

Thanks, these are very encouraging comments!

Weaknesses:

- The method is only tested on a specific sort of RBNs (Sec. G, 1). What about having different rules?

In our tests on RBNs, we varied two main parameters: (1) the average degree of the networks and (2) the bias on the output of the lookup tables. Changing parameter (1) allows us to explore both the ordered and chaotic regimes of Boolean dynamics. Tuning the value of parameter (2) provides us with an immediate way of controlling for the type of Boolean rules that regulate the dynamics of the network. We recognize that is not a comprehensive numerical study for RBNs. Specifically, we consider homogeneous and uncorrelated networks only. However, uncorrelated and unbiased networks in the chaotic regime should be among the most difficult types of Boolean networks to control. In this sense, they provide an upper bound for the size of the minimal driver set required to control the behavior of other networks with similar size and average degree. This was the main purpose of our systematic analysis on RBNs. We slightly revised the text to make it apparent.

- The paper is generally not well-structured. Directly after introduction the results come and one needs to go the last section for the method, formulas and details ...

We used a structure that adheres with the journal's requirements, see <https://www.nature.com/documents/ncomms-submission-guide.pdf>.

Comments and questions:

- the title and abstract are not reflecting well the idea of the paper

Thanks for the comment. We revised the abstract to better distill the content and main message of the paper. We kept the title unchanged. We believe the title succinctly summarizes the main idea behind the methods developed in the paper, i.e., the generalization of influence maximization from spreading processes to Boolean dynamical systems.

- having some schematic figures with some examples can help a lot to understand the idea easier, either in introduction or method section.

We already have an illustration of a specific Boolean network, see Figure 3 of the paper. The figure should deliver an intuitive idea of how we generalized the concept of influence maximization from spreading processes to Boolean dynamics. Also, we revised the description of the greedy optimization algorithm (see Methods section) to properly highlight the various steps used in our optimization method. We hope that these revisions address the criticism by the reviewer. If not, we would very happy to receive additional directives from the reviewer to address properly the criticism.

- discussing about some limit cases would help a lot. For instance, when dynamics have only one attractor or when number of attractors are too high but they are small, or when you choose some specific x set (topologically)...

Thank you for the observation! We expanded the conclusions, listing some of the potential limitations that may affect our proposed method.

- are x sets chosen from specific sub-graphs in the network? or have you checked if the optimised sets have any specific topological feature or position in the network?

Thank you so much for the great suggestion! We performed a new analysis where we studied the conditional probability of a node to be in the minimal driver set depending on its in-/out-degree. Results are reported in the revised version of the SI, and commented in the main paper. To facilitate the assessment by the reviewer, we display the results also in Fig. R9.

As expected, we find that input nodes are always part of the driver set. These nodes by definition cannot be controlled if not via an external input. Nodes with sufficiently large in- and out-degree are significantly more likely to be in the minimal driver set than nodes with small in-/ out-degree centrality. This fact indicates that topologically central nodes are likely to be part of the minimal driver set.

Our results are far from being conclusive. However, they nicely provide a structural interpretation of our dynamical analysis. We thank the reviewer for the suggestion.

Figure R9: Topological centrality of nodes in the minimal driver sets. Results are for nodes in all 74 networks in the Cell Collective repository. (a) Each node is represented by its in- and out-degree. We display all nodes from all networks of the repository. We color points depending on whether the node is identified within the minimal driver set by the unconstrained greedy selection algorithm. (b) Conditional probability of a node to be part of the minimal driver set depending on its in-degree. We excluded from the analysis all nodes with null in-degree as they are always identified as drivers. The dashed blue line indicates the expected probability with no dependence on the in-degree; error bars indicate one standard deviation away from the expected value in such a null model. (c) Conditional probability of a node to be part of the minimal driver set depending on its out-degree. Reference value and error bars are obtained in the same way as for panel b.

- how good does this method work in different (ordered-disordered) regimes (see this paper for instance: PRL 107 (18), 188701, 2011) one can argue by choice of boolean rules and how perturbing the x sets ...

Our analysis on the RBNs indeed spans the range from the ordered to the chaotic regimes: RBNs with degree $k = 1$ are in the ordered regime; RBNs with degree $k \geq 2$ are in the chaotic regime. Networks we considered in the ordered regime do not require control as suggested by Figure 1a (they quickly converge to a steady state). Chaotic RBNs instead are difficult to control, as the results of Figure 4a show (control becomes increasingly difficult as k increases). RBNs with $k \geq 2$ become easier to control as the bias in the dynamical rules increases, see Figure S10. Increasing the bias pushes the network towards the ordered regime.

- I missed the point how the desired attractor is chosen.

We slightly rephrased the text to clarify the issue. Our method does not prescribe how to choose an attractor. Rather, it aims at identifying, via constrained optimization, the minimal driver set toward a pre-specified attractor. In some networks, attractors may not be known in advance; our unconstrained optimization method is still useful to find an optimal driver set toward an unspecified attractor (generally the one with the largest basin of attraction).

- the algorithm, e.g. flowchart can be included in the supplementary material.

We expanded the Methods section to account for the comment made by the reviewer. We itemized the various steps used in the algorithm for the identification of optimal driver sets. The description is on purpose kept brief. To avoid repetitions, we did not include any flow charts in the SI.

We stress that we rely on a standard greedy strategy, used quite often in the solution of discrete optimization problems.

- in Fig 3. b, do all the 10 attractors have size 1?

Yes, these are fixed points of the dynamical system.

List of changes

To ease the reassessment by the reviewers, major changes in the revised manuscript have been highlighted with red fonts. This is the list of major changes that we performed:

1. We revised the abstract as requested by reviewer 3.
2. We expanded the introduction to account for the comments by reviewer 2 about existing methods for the identification of driver sets in Boolean networks.
3. We expanded the introduction to summarize the results obtained under various updating schemes. This novel analysis was prompted by comments of referee 1.
4. We incorporated multiple new paragraphs in Secs. II A-C to describe novel validation tests (indicated by reviewer 2) and additional results valid for updating schemes other than synchronous updating (as suggested by reviewer 1). All these new results are documented in the revised version of the SI. We also revised Figure 4 to include a direct comparison between the predictions of our method and those by Borriello and Daniels's method (Fig. 4d).
5. Conclusions have been greatly expanded as recommended by reviewer 3.
6. Revisions have been performed also in several parts of the Methods section to address comments by reviewers 2 and 3.
7. We included 8 new figures and 1 new table in the revised version of the SI. Some of the old figures have been revised. We avoided the use of red fonts to highlight changes in the SI given the extensive amount of new material appearing in the revised version of the document.

Bibliography

- [1] Eric Goles, Marco Montalva, and Gonzalo A Ruz. Deconstruction and dynamical robustness of regulatory networks: application to the yeast cell cycle networks. *Bulletin of mathematical biology*, 75(6):939–966, 2013.
- [2] Enrico Borriello and Bryan C Daniels. The basis of easy controllability in boolean networks. *Nature communications*, 12(1):1–15, 2021.

REVIEWERS' COMMENTS

Reviewer #1 (Remarks to the Author):

The authors have conducted new simulations based on my previous comments. They have addressed the main issues satisfactorily. I believe the work is now publishable.

Reviewer #2 (Remarks to the Author):

The authors have made significant additional analyses that convince me of the validity and usefulness of the results. I am happy to recommend publication.

Reviewer #3 (Remarks to the Author):

The revised version is well improved. The authors responded well the questions and comments too. It can be published.

First referee

The authors have conducted new simulations based on my previous comments. They have addressed the main issues satisfactorily. I believe the work is now publishable.

Second referee

The authors have made significant additional analyses that convince me of the validity and usefulness of the results. I am happy to recommend publication.

Third referee

The revised version is well improved. The authors responded well the questions and comments too. It can be published.

Our reply to the reports

We thank all reviewers for the time dedicated to our manuscript. We appreciate the very positive reports.